# Powertrain Modal Analysis for Defining the Requirements for a Vehicle Drivability Study

**Federico Longoni [1], Anders Hägglund [1], Francesco Ripamonti [2],\* and Paolo L. M. Pennacchi [2]**

1 Volvo Car Corporation, 418 78 Gothenburg, Sweden
2 Department of Mechanical Engineering, Politecnico di Milano, 20156 Milan, Italy
\* Correspondence: francesco.ripamonti@polimi.it

**Abstract:** The powertrain of a car plays a major role in establishing the vehicle's offered comfort due to vibrations because it is the heaviest single component installed on the chassis; therefore, when oscillating, it transmits considerable forces to the chassis, inducing unwanted vibrations. For this reason, it is important to identify some associated properties with the powertrain suspension system that describe the performance of its rigid body dynamics. In this way, we could place constraints and requirements on these quantities in order to exclude all the configurations that cause intolerable levels of vibrations, and include all the others in the analysis for further evaluation. The definition of these requirements is critical: a poor setting of requirements excludes potentially good powertrain suspension setups and includes those ones with a drivability index that is too poor. In this paper, we identify a set of quantities that show correlation with the vibration performance of the powertrain setup. A method for testing the specificity of the requirements is also shown in order to evaluate which requirements perform best when it comes to filtering engine suspension setups that provide an acceptable level of vibrations.

**Keywords:** drivability; engine suspension setup; modal quantities; modal energies; engine mounts; requirement specificity





## 1. Introduction

Comfort is a keystone in a vehicle's success in the modern automotive industry; in particular, an increasing number of automotive companies are setting up a separate department for noise and vibration harshness (NVH) because it is something that the customer can experience much more easily and complain about. NVH comfort depends on many variables, such as high- and low-frequency vibrations, noise, and handling [1–5]. Since it is difficult to quantify NVH altogether, one possible approach is to combine the aforementioned variables in order to formulate a vehicle setup capable of describing some aspects of comfort performance. Among these attributes, drivability is influenced by both objective measurements and subjective impressions that the driver receives [6–8]. Subjective drivability is significantly deteriorated by longitudinal vibrations, since the vehicle is not constrained in that direction, causing the whole chassis to vibrate if excited in this sensitive direction.

Since the powertrain of a vehicle accounts for 10–15% of the total mass, car manufacturers pay attention to the way in which it is attached to the chassis, because if too-high an oscillation value is allowed, greater forces are transmitted, inducing vibrations. The powertrain is attached to the chassis by means of engine mounts; their properties (stiffness, position, and damping) determine the powertrain's rigid body dynamics and thereby the extent of vibrations transmitted to the chassis. Thus, the chosen engine mount setup is directly linked with the vehicle's drivability performance. Engine suspension designers spend time on finding the best engine mount properties that allow for a lower transmission of forces. This optimisation process can take advantage of analytical considerations and numerical simulations to evaluate the performance of an engine's suspension setup [9–16].

In the literature, different approaches have been adopted, in order to optimise this index, and powertrain suspension setups with fully decoupled eigenmodes generally perform well. A fully decoupled eigenmode is a modal shape that involves only one direction of vibration at a time. Despite that fact that it has not yet been demonstrated that fully decoupled eigenmodes are a necessary condition for the highest comfort performance, decoupled powertrain configurations provide better NVH performance [17–21].

Due to the difficulty in achieving ideal decoupled suspension setups at the design stage, it is useful to identify quantities on which to base requirements in order to discriminate between the powertrain suspension setups that are worth considering from the others. Different techniques have been developed to tune engine mount properties and approach these configurations. These techniques, such as the torque roll axis (TRA) or the elastic axis (EA) [11,17,22], also provide insight into the quantities that characterise good engine mount setups.

As an alternative, it is also possible to formulate the requirements using the modal quantities that describe how far we are from the ideal fully decoupled condition. The innovative element introduced in this paper lies in the identification of new modal quantities that influence the longitudinal vibration transmission problem, and the evaluation of the requirement's performance in terms of design improvements. In this paper, the correlation between modal quantities (or more generally quantities) and the longitudinal vibration transmission is analysed in order to allow for the better formulation of the requirements that discriminate between engine mount setups that transmit a low value of vibration in the longitudinal direction from the others. This facilitates identifying the tuning target that needs to be modified in order to establish engine mount setups that transmit lesser forces in the longitudinal direction.

## 2. Formulating the Problem

In the literature, the concept of modal kinetic energy is widely used as a way to quantify the combination of degrees of freedom within a modal shape [22,23]. The popularity of these quantities for describing how well the powertrain suspension setup performs is due to their meaning. Modal couplings represent the percentage contribution to the response in a given mode for each degree of freedom (DOF). Under the assumption that the ideal suspension setup is fully decoupled, the designer tries to minimise more than one modal couplings within the same modal shape in order to decrease the interaction between the input excitation direction and the critical directions (the longitudinal direction in this case).

The formulation usually adopted to compute these modal couplings is that given in many scientific papers [23]. Less popular is the formulation based on the idea of the mode in the participation factor state. The concept of the modal participation factor for mechanical system dynamics was investigated by Abed et al. [24–28] and can be formalised as follows:

$$p_{ji} := \phi_{ij}^L \phi_{ji}^R \tag{1}$$

where $p_{ji}$ is the participation factor of the *i*-th mode into the *j*-th state, and $\phi^L$, $\phi^R$ are the left and right eigenvectors, respectively, defined such that $[\Phi^R][\Phi^L] = [\delta_{ij}]$, where $\delta_{ij}$ is the Kronecker delta. Matrices $[\Phi^R]$ and $[\Phi^L]$ are assembled as follows:

$$[\Phi^R] = \begin{bmatrix} \underline{\phi_1^R} & \underline{\phi_2^R} & \cdots & \underline{\phi_i^R} \end{bmatrix} \tag{2}$$

$$[\Phi^L] = [\Phi^R]^{-1} = \begin{bmatrix} \underline{\phi_1^L}^T \\ \underline{\phi_2^L}^T \\ \vdots \\ \underline{\phi_i^L}^T \end{bmatrix} \tag{3}$$

The definition of the modal participation factor of mode *i* in state *j* refers to a condition in which the excited system state is the *j*-th one, as indicated by [25]. Due to the neater formulation and more thorough mathematical derivation in the literature, compared to [22,23], modal couplings are calculated using the participation factor convention.

The participation factor can be interpreted as a fraction that represents the amount of excitation energy that excites/triggers the *i*-th modal shape of the system. These fractions are organised within a matrix that is called the participation factor matrix. We may refer to both modal coupling as participation factors and vice versa since they provide the same results. Below, when we refer to a modal coupling (modal coupling matrix) we name both the mode and the direction to which it refers (see participation factor matrix in Table 1 and the reference system shown in Figure 1). An example of a participation factor matrix is shown in Table 1, where it is possible to see how an excitation along the *TX* direction excites *TX* mode by 67.47%, and *TZ* mode by 18.95%. Conversely, reading the figures in the table columns, it is possible to understand the relative importance of each DOF to the corresponding mode in terms of the participation factor.

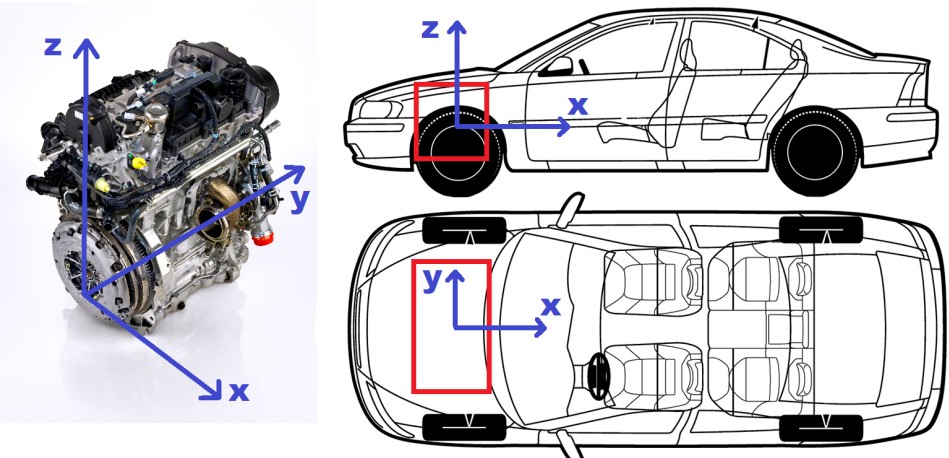

**Figure 1.** Powertrain and vehicle coordinate systems. The red rectangle represents the powertrain's position within the vehicle.

Therefore, thanks to modal analysis theory, it is possible to derive the time domain contribution of each modal shape to the free total response of the mechanical system. The theoretical derivation mentioned below references Abed et al. [25]. The modal decomposition of the time response can be formulated as follows:

$$\underline{x}(t) = \sum_{i=1}^{n} (\underline{\phi}_i^L \underline{z}_0) e^{\lambda_i t} \underline{\phi}_i^R \tag{4}$$

In this way, it is possible to highlight the contribution of the *i*-th modal shape to the *k*-th output response included in the $\underline{x}$ vector. For instance, let us consider a three-degrees-of-freedom system. It is possible to define the vector that contains the initial displacement conditions as $\underline{x}_0 = \begin{bmatrix} K_1 & K_2 & K_3 \end{bmatrix}^T$, and rewrite the response $\underline{x}(t)$ as

$$\underline{x}(t) = \sum (\phi_{i1}^L K_1 + \phi_{i2}^L K_2 + \phi_{i3}^L K_3) e^{\lambda t} \begin{bmatrix} \phi_{1i}^R \\ \phi_{2i}^R \\ \phi_{3i}^R \end{bmatrix} \tag{5}$$

**Table 1.** Example of participation factor matrix. The sum of the coupling percentage columns is 100% when using the participation factor formulation.

| Participation Factor Matrix (%) | | | | | | | |
|---|---|---|---|---|---|---|---|
| DOF | TX-Mode | TY-Mode | TZ-Mode | RX-Mode | RY-Mode | RZ-Mode | Sum |
| TX | 67.47 | 2.72 | 18.95 | 0.46 | 0.47 | 9.93 | 100% |
| TY | 5.18 | 92.94 | 0.12 | 1.11 | 0.39 | 0.26 | 100% |
| TZ | 14.01 | 0.75 | 64.96 | 0.01 | 20.27 | 0.0 | 100% |
| RX | 0.0 | 1.45 | 0.23 | 94.91 | 0.67 | 2.74 | 100% |
| RY | 4.30 | 1.07 | 14.76 | 0.64 | 77.79 | 1.44 | 100% |
| RZ | 9.04 | 1.07 | 0.98 | 2.87 | 0.41 | 85.63 | 100% |
| Sum | 100% | 100% | 100% | 100% | 100% | 100% | |

In the case of an initial state condition corresponding to term $K_1$ in vector $\underline{x}_0$ ($K_2 = 0$ and $K_3 = 0$), which corresponds to an excitation given only in one direction, we can expand the summation for $i$ going from 1 to 3, obtaining

$$\underline{x}(t) = (\phi_{11}^L K_1 e^{\lambda_1 t} \begin{bmatrix} \phi_{11}^R \\ \phi_{21}^R \\ \phi_{31}^R \end{bmatrix} + \phi_{21}^L K_1 e^{\lambda_2 t} \begin{bmatrix} \phi_{12}^R \\ \phi_{22}^R \\ \phi_{32}^R \end{bmatrix} + \phi_{31}^L K_1 e^{\lambda_3 t} \begin{bmatrix} \phi_{13}^R \\ \phi_{23}^R \\ \phi_{33}^R \end{bmatrix}) \quad (6)$$

The eigenmodes, each multiplied by the relative exponential for the corresponding eigenvalue $\lambda_i$, are shown between brackets. The left eigenvector terms multiplied by initial condition $K_1$ represent the amount of energy introduced in each eigenmode. It follows that the extent of the participation of each mode in the final response depends on the initial conditions given by the left eigenvectors.

The theoretical derivation described above can indicate new modal quantities that describe the participation of each modal shape in the state response of the dynamical system when a certain initial condition is applied. In particular, on the basis of the assumption of the linear dynamics of a system, the superposition principle holds; thus, any kind of initial condition can be decomposed into fundamental ones applied to a single state. Considering only one initial condition of a single state at a time, we end up with a matrix of new modal quantities that can describe the interaction and contribution of the modes to the final response. Equation (6) can be rearranged into a matrix form highlighting a new modal quantity matrix:

$$\underline{x}(t) = K_1 \begin{bmatrix} \phi_{11}^L \phi_{11}^R & \phi_{21}^L \phi_{12}^R & \phi_{31}^L \phi_{13}^R \\ \phi_{11}^L \phi_{21}^R & \phi_{21}^L \phi_{22}^R & \phi_{31}^L \phi_{23}^R \\ \phi_{11}^L \phi_{31}^R & \phi_{21}^L \phi_{32}^R & \phi_{31}^L \phi_{33}^R \end{bmatrix} \begin{bmatrix} e^{\lambda_1 t} \\ e^{\lambda_2 t} \\ e^{\lambda_3 t} \end{bmatrix} \quad (7)$$

A synthetic expression for the new modal quantities contained in the matrix of Equation (7) is

$$\Gamma_{ij} = \phi_{ik}^L \phi_{ji}^R \quad (8)$$

where $i$ indicates the eigenmode under consideration, $j$ is the output direction to which the modal quantity refers, and $k$ is the state to which the initial condition is applied. Indices $i$, $j$, and $k$ can have a value between 1 and $n$, where $n$ is the number of degrees of freedom of the considered system.

Since there are three subscripts in the definition of the new modal quantities ($\Gamma_{ij} = \phi_{ik}^L \phi_{ji}^R$, i.e., $i$, $j$, $k$), a 3D matrix is needed to represent all the terms contained in $\Gamma$; Figure 2 shows matrix $\Gamma$ in its full form.

The procedure derived above refers to the work by Abed [25], despite the fact that the only quantities considered for further analysis in that work were those with $k = j$, reducing the analysis to the quantities contained in the corresponding 2D matrix. In this paper, the influence of the other, previously not investigated, quantities is studied for the purpose of the transmission of longitudinal vibrations from a vehicle's powertrain to its chassis.

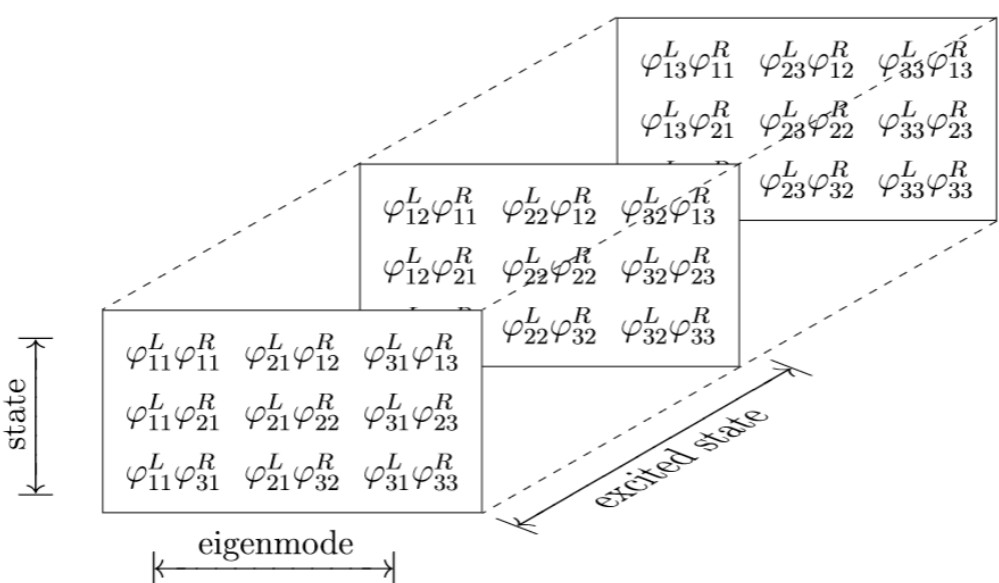

**Figure 2.** Three-dimensional representation of new modal quantities matrix $\gamma$ in the case of a 3-degrees-of-freedom system (3DoF).

## 3. Design of the Experiments

In order to investigate the possible relationships between the quantities and the extent of longitudinal vibrations transmitted by the powertrain to the chassis, a metric is needed.

To evaluate the performance of a suspension configuration, the behaviour of the system when subjected to excitations must be analysed. It is not possible to draw any conclusion a priori about the chosen configuration by looking at only a physical or modal parameter without information about how they correlate with the time domain behaviour of the powertrain suspension setup. For this reason, it is necessary to simulate the dynamics of the configuration of the mounts when the powertrain is excited. Considering the 3D case, the simulation takes into account an input excitation given as an impulse torque about the $y$ axis (usual torque supplied by the engine to the driveline).

The used metric is the vibration dosage value [29] with a slight modification in its formulation because it considers the forces transmitted in the longitudinal direction by the powertrain instead of the acceleration. It is formulated as follows:

$$\text{VDV} = \sqrt[4]{\int_{t_i}^{t_f} F_x^4(t)\,dt} \tag{9}$$

where $t_i$ is the start time of exposure, $t_f$ is the final time of exposure, and $F_x$ is the resultant force transmitted in the longitudinal direction ($x$ direction) by the powertrain to the chassis. The VDV, whose unit of measurement is $[Ns^{\frac{1}{4}}]$, makes it possible to consider the transient behaviour of the powertrain suspension system. The VDV formulated above is a cumulative measure of the transmitted force over time; therefore, higher values of VDV correspond to higher integral values of the transmitted vibration along the specified direction.

The powertrain suspension setup was modelled as a 6DOF rigid body connected to the ground (chassis) by means of engine mounts characterised by stiffness and damping in the three directions (see Figure 3). All the physical properties and the dynamics of the mechanical system were considered under the assumption of linearity. The inertial characteristics of the considered powertrain were such that its mass was 275 kg, and its rotational inertia was approximately equal to $I_{xx} = 20$ [kg/m$^2$], $I_{yy} = 10$ [kg/m$^2$], $I_{zz} = 17$ [kg/m$^2$]. As said previously, the input excitation considers only the impulse torque given about the $Y$ axis. Thanks to the impulse theorem, this excitation is schematised as the initial condition given for the corresponding state (rotation about the $Y$ axis). Due to the linearity assumptions,

the magnitude of the excitation can be rescaled according to our needs, and the output is scaled too.

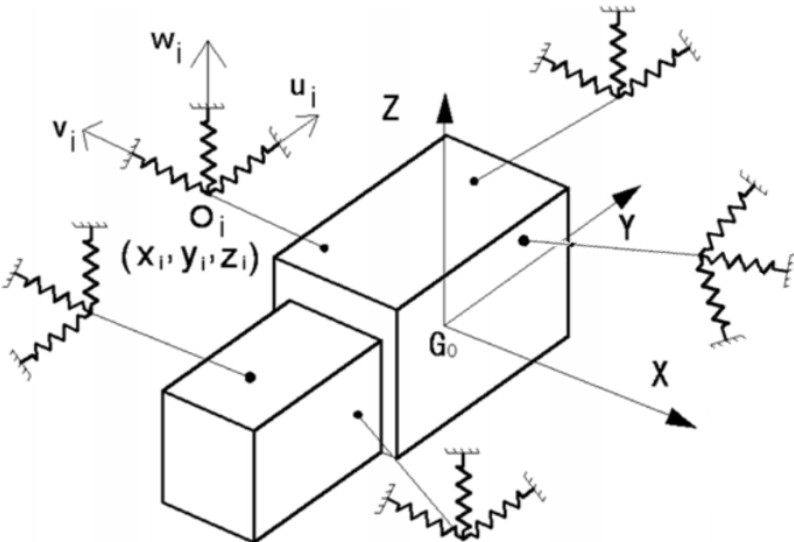

**Figure 3.** Six-DOF powertrain model.

Analysis based on the Monte Carlo method was carried out by randomly sampling the physical parameters associated with the modelled powertrain suspension setup within the range of ±50% in relation to their nominal value. In this way, it is possible to compute the modal properties associated with the investigated engine mount setup, and, after simulating their dynamics, to record the VDV performance. Eventually, the correlation between VDV behaviour and selected modal quantities can be investigated. In order to have more reliable results, it would be better to have denser scatter points and an even distribution (see Figure 4).

Unfortunately, a drawback associated with this approach leads to indirect control over the modal quantities, raising issues not only with respect to the range of inspected values, but also from a density point of view. In order to mitigate this issue and to better understand the influence that modal quantities have on powertrain performance when there is coupling between modal shapes, the number of the initial powertrain suspension setups taken into consideration was extended significantly.

The results were processed to identify trends between the quantities and VDV performance of the powertrain suspension setups. A trend means a correlation between the two quantities; therefore, by limiting the value of one, the other is influenced too. Thus, it is possible to discriminate between suspension setups that correspond to low values of VDV from the others. Scatterplots were used to investigate these trends/correlations, where each point in a scatterplot corresponds to an engine mount setup that was simulated. This setup could be uniquely described in a multidimensional hyperspace, but the scatterplots with the colour scale corresponding to the VDV level allow for a better analysis of the problem.

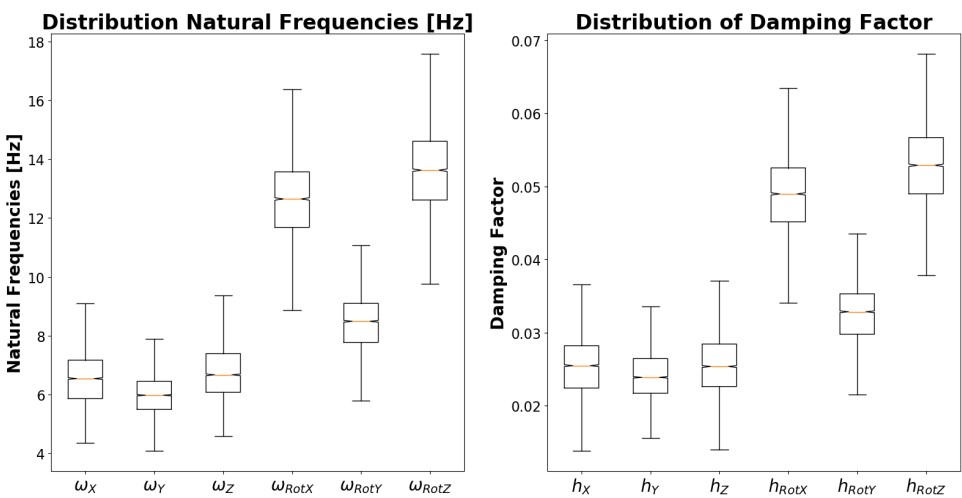

**Figure 4.** Range of investigated natural frequencies and damping factors.

The scatterplots shown in Figure 5 are representative of the most significant combination of quantities that can separate powertrain suspension setups that provide a low VDV to the chassis from those that transmit high vibration levels. Indeed, the selected scatterplots facilitate separating the blue and red points by properly selecting the requirements. On the *Y* axis, the distance of the elastic axis from the centre of gravity in the vertical direction (EA-CG z dir) is always displayed. This quantity is known in the literature to correlate well with the longitudinal vibration transmitted by the powertrain to the chassis; therefore, it is included in all the scatterplots. On the *X* axis, instead, some quantities showing the best correlation were selected. The impurity of a mode represents how far the mode is from a perfect fully decoupled condition. The lower the impurity is, the more the mode involves only the direction after which it is named. The other coupling quantities represent the kinetic percentage energy associated with a particular direction within a modal shape. Lastly, the quantities displayed on the right-hand side of Figure 5 represent the new modal quantities introduced in the previous section.

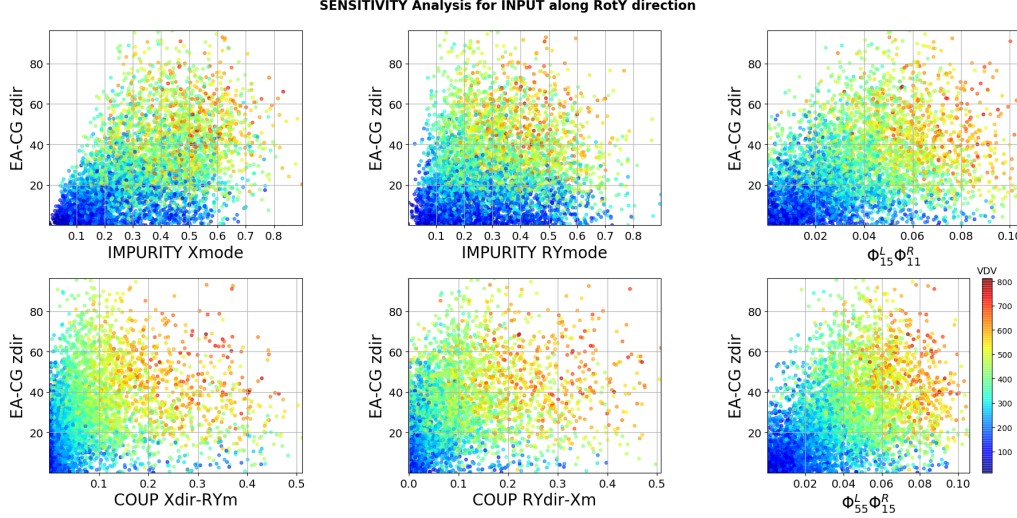

**Figure 5.** Scatterplots showing how the chosen modal quantities can be combined with the vertical distance of the elastic axis (EA) from the centre of gravity (CG).

## 4. Requirement Definition and Performance Analysis

To filter the engine suspension setups that correspond to a low VDV value from those that transmit higher vibration levels, a proper formulation of requirements is needed. There

are many possible combinations of variable constraints that allow for different selections of engine suspension setups. Among them, this paper focuses on the comparison between two requirements formulated as follows:

- Two inequalities using the modal coupling information only, which constitutes the requirements guidelines currently used by most car manufacturers (benchmark requirement).
- An inequality line inequality, including the new modal quantity $\Phi_{55}^L \Phi_{15}^R$ (new requirement).

These requirements are shown in Figure 6.

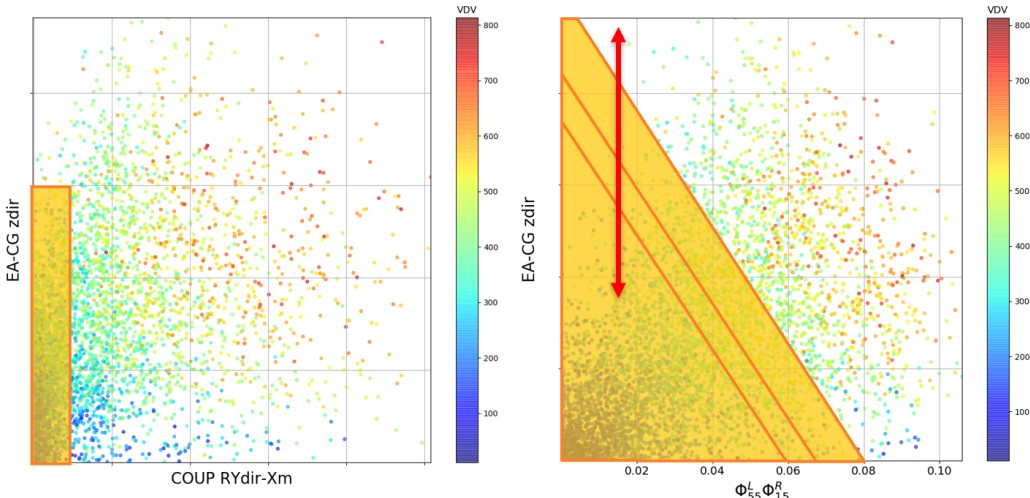

**Figure 6.** Comparison of the formulation of different requirements: (**left**) benchmark requirement; (**right**) new requirement.

The inequalities that define the first requirement are formulated as two upper limits, one on the EA-CG distance and one on the modal coupling of the $X$ mode in the rotation $Y$ direction. For confidential reasons, no specific value is given. Instead, the inequality that defines the second requirement is

$$\text{EA-CG}_{Zdir} \leq \mathbf{m}\phi_{55}^L \phi_{15}^R + \mathbf{q} \tag{10}$$

because the VDV seemed to increase along an oblique line with positive slope. Therefore, if the requirement is set to describe the equation of a line, it may be possible to improve the number of acceptable suspension configurations, excluding just a small number of them. Slope $\mathbf{m}$ was tuned manually according to the steepest ascent direction of the VDV, and it was set to $-\frac{100}{0.08} = -1250$. The intercept on the $Y$ axis was tuned in order to compare the two requirements in similar conditions.

The analysis of an ideal requirement formulation would include all powertrain suspension systems that complied with a VDV value, and exclude all the setups that provided intolerable levels of the transmitted forces in the $X$ direction.

Figure 7 shows how the investigated dataset was filtered according to the benchmark requirement. The histogram describes the number of engine mount setups that produce the corresponding level of VDV shown on the horizontal axis. The light blue area in the background represents the full dataset investigated by the DOE. The green area represents the number of points that satisfy the requirements applied to the dataset. Lastly, the red area corresponds to the points that did not satisfy the requirement's condition. The green and red areas were complementary to the light blue area. Ideally, the perfect requirement would be able to completely separate engine mount setups with a VDV lower than the desired threshold from the others, i.e., it would be able to fully separate the green and red areas.

In order to be able to distinguish between acceptable and unacceptable configurations, a threshold level is needed to compare the performance results. The two VDV threshold

values considered in this paper are represented as yellow and black lines, as specified in the legend.

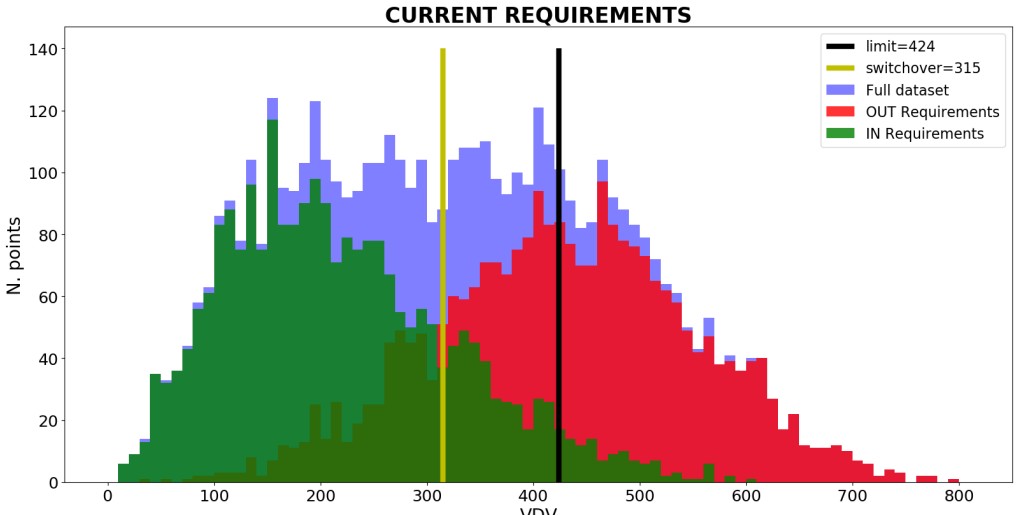

**Figure 7.** Histogram showing the separation of VDV performance for the engine mount setup.

The first threshold (black) was set to VDV = 424. This value was derived from the assumption that, according to experience, when applying the benchmark requirement, provided a tolerable VDV level; therefore, the threshold was chosen as the 95th percentile of the VDVs included in the analysis by the benchmark requirements.

The definition of an absolute limit for the VDV makes it possible to evaluate the effectiveness and efficiency of the requirement's specificity for including or excluding setups in the analysis, but the requirement's performance can also be evaluated in terms of the transitional speed from acceptable to unacceptable configurations. Therefore, we can define a second threshold (yellow) called the switch-over line that occurs at a VDV for which the number of excluded acceptable setups is the same as the number of included unacceptable ones.

Therefore, the black line represents an absolute value chosen according to experience, while the yellow line is tuned according to the kind of requirement under consideration, and we can say that it is a relative limit.

Figure 8 shows the specificity of the requirements when the limit threshold VDV is taken into consideration. Visually speaking, it is possible to see an improvement in the number of acceptable engine mount candidates that can be considered for further analysis when the new requirement is applied ($q = 95.7$). Similarly, Figure 9 shows the comparison between the histograms when the switch-over line is considered. In order to compare the two requirements in relation to the same switch-over line, the **q** value was set to 67.3.

In this case, there was also an increment of potentially good setups included in the analysis, i.e., the green area was increased. In order to compare the performance of different requirements, it is possible to define a synthetic performance index related to the threshold value considered (limit or switch-over line) as follows:

$$\text{RPI} = \frac{\text{AT-I set-ups}}{\text{BT set-ups}} + \frac{\text{BT-E set-ups}}{\text{BT set-ups}} \tag{11}$$

where RPI stands for the requirement performance index, AT-I setups are the above-threshold but included setups, BT-E setups are the setups below the threshold but excluded from the analysis. Lastly, BT setups are representative of all the setups below the considered threshold.

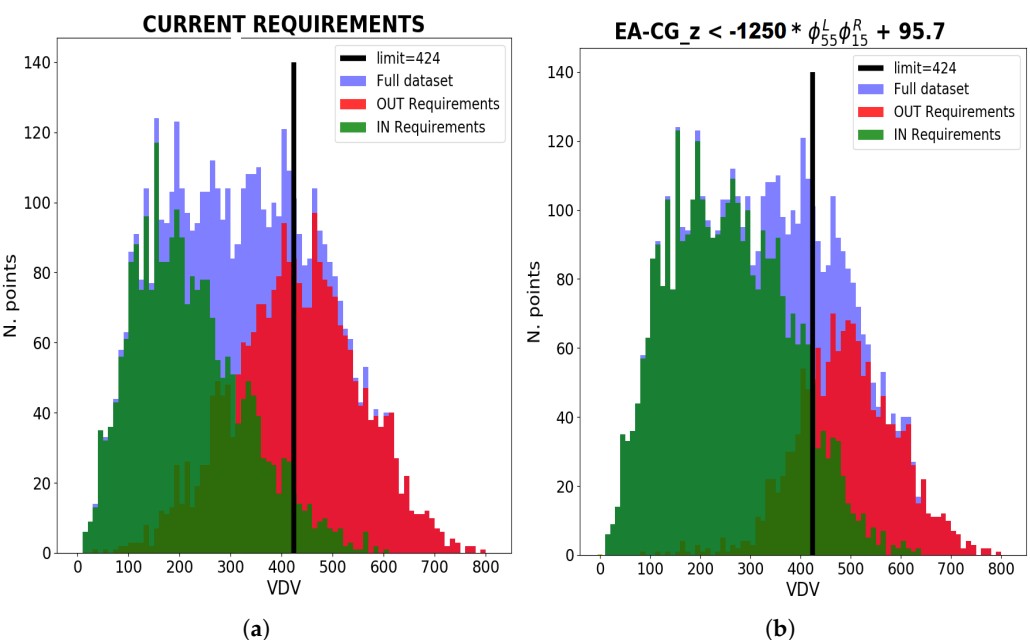

**Figure 8.** Specificity tests for the (**a**) benchmark and (**b**) new requirements with $q = 95.7$ in relation to the limit-VDV.

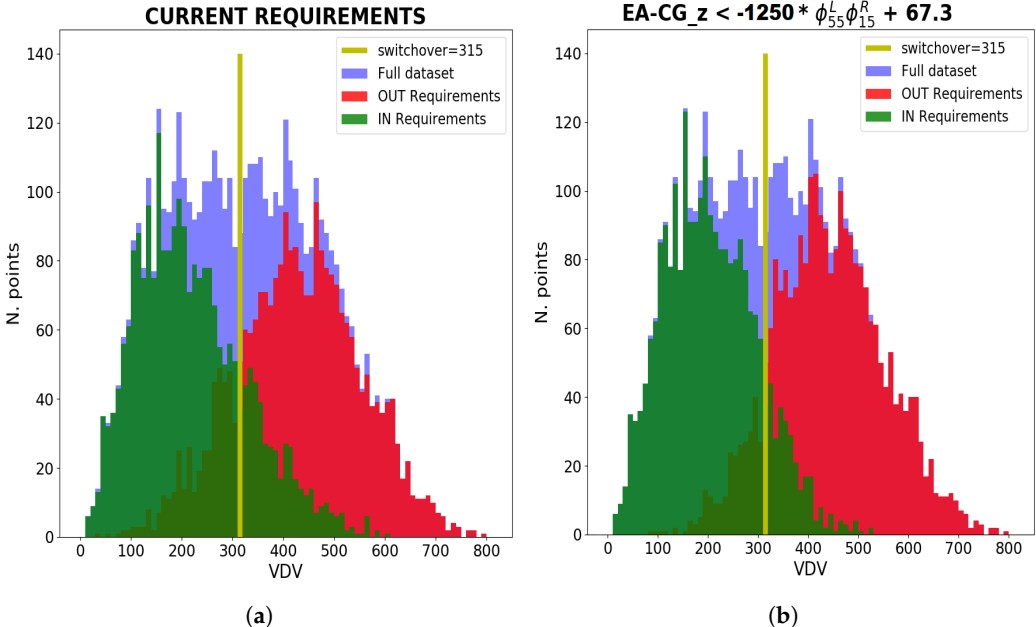

**Figure 9.** Specificity tests for (**a**) the benchmark and t (**b**) new requirements with $q = 67.3$ in relation to the switch-over VDV.

The results for the two requirements are shown in Table 2, showing an improvement in the number of powertrain suspension setups included in the analysis by adopting a requirement definition that uses the new modal quantities. This is due to a decrease in the value of the requirement performance indices, meaning that we were closer to the ideal condition where only and all acceptable powertrain suspension setups were included in the analysis.

**Table 2.** Comparison of requirement indices between the benchmark and new formulations.

|  | Benchmark Requirements | New Requirements |
|---|---|---|
| Limit requirement index | 0.3841 | 0.1882 |
| Switch-over requirement index | 0.3858 | 0.2477 |

## 5. Conclusions

In this paper, a new approach for the definition and comparison of requirements for the powertrain suspension setup evaluation process was proposed. New modal quantities, the terms in the participation factor matrix, correlated quite well with the time domain response of the powertrain modelled (in terms of VDV). The formulation of the requirements, including the new modal quantities, gave rise to an improvement in the performance index, as shown in Table 2, where lower performance index values were associated to the new requirements. For the considered test case, the new modal quantities better described the coupling between modal shapes and degrees of freedom.

As the main outcome of this approach, the implementation of the proposed requirements could speed up the design process, limiting detailed simulations to only the most promising configurations. Moreover, the formulation of the requirements could help a designer in identifying the quantities that are most responsible for the transmission of vibration from the powertrain to the chassis. The formulation of the modal requirements presented in this paper could also help in the definition of objective functions in optimisation processes during the design phase of powertrain suspension setups.

**Author Contributions:** Conceptualization, F.L., A.H., F.R. and P.L.M.P.; methodology, F.L., A.H. and F.R.; software, F.L. and A.H.; validation, F.L., A.H. and F.R.; formal analysis, F.L., A.H., F.R. and P.L.M.P.; investigation, F.L. and A.H.; resources, F.L.; data curation, F.L.; writing—original draft preparation, F.L.; writing—review and editing, F.L., A.H. and F.R.; visualization, F.L.; supervision, A.H., F.R. and P.L.M.P.; project administration, A.H. and F.R.; funding acquisition, F.R. and P.L.M.P. All authors have read and agreed to the published version of the manuscript.

**Funding:** This research received no external funding.

**Institutional Review Board Statement:** Not applicable.

**Informed Consent Statement:** Not applicable.

**Data Availability Statement:** The reference data presented in this study is available on request from the corresponding author.

**Conflicts of Interest:** The authors declare no conflict of interest.

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
