# Peer review of "Powertrain Modal Analysis for Defining the Requirements for a Vehicle Drivability Study"

_machines, doi:10.3390/machines10121120_

Round 1
Reviewer 1 Report
Abstract, fist sentence, typing mistake: "vejhicle's"
Figure 1 is in Introduction. And its first mention is in next page line 89. It should be first mention and than presented figure. Here should be Figure 1 after line 89. Same for all Figures.
Table 1 is given without some explanations. Please give us more informations about it.
Authors gave methodology how to test powertrain in the car for the aspect of vibrations. From my point of view it can be useful in vehicle industry.
This is good example how something should be tested. Maybe this paper can not be classified as Article and maybe is better to be Technical Note.
Conclusions are short and clear. Maybe it can be expanded.
References are ok, it could be better but this level is acceptable.
Author Response
We thank the Editor and the Reviewers for the comments on our work. We have studied them carefully and addressed in this letter. The changes in the manuscript have been highlighted in red. We hope that this revised work and our responses to the questions are satisfactory to the reviewers.
Replies to reviewer 1:
- Typo in the abstract has been corrected
- The figures’ position has been updated in order to have all figures and tables shown after they are cited in the text.
- More context about Tab.1 is given. An explanation on how to read it and the meaning of the figures within it has been introduced. The changes are highlighted at pag. 3 of the revised manuscript.
- Conclusions have been extended.
Reviewer 2 Report
This research proposed a definition of requirements for powertrain driveability study. This article is very interesting to read.
In this research, the judgement of suspension quality was based on the vibration dosage value. The reader could not clearly understand the physical differences between high and low VDC value.
The authors did not include powertrain models in the paper. It is difficult to understand the improvement of the driveability based on the VDC value. Authors should provide the connections between the VDC and driveability.
Since there was no powertrain model, the frequency and mode shape of powertrain were not presented in the paper. Readers have difficulty to know the direction to improve the powertrain driveability.
Author Response
We thank the Editor and the Reviewers for the comments on our work. We have studied them carefully and addressed in this letter. The changes in the manuscript have been highlighted in red. We hope that this revised work and our responses to the questions are satisfactory to the reviewers.
Replies to reviewer 2:
- A brief description on how to interpret/compare different Vibration Dosage Values has been introduced at pag. 5 of the revised manuscript.
- As introduced at line 24, drivability depends on both objective measurements and subjective impressions. If we adopted such a definition of drivability, we would introduce a significant variability in the results interpretation. Instead, since longitudinal vibrations play a major role in deteriorating drivability, we decided to limit the analysis to the comparison of different VDV level. Starting from this point we can focus on the requirements formulation.
- The powertrain model was not included in the paper because the aim of the work focuses more on the methodology to create new requirements on modal quantities rather than specific configurations. The benchmark requirement, used as reference case, is a guideline (as stated at line 205 of the manuscript) and does not refer to a specific set of powertrains.
Round 2
Reviewer 2 Report
This research proposed a definition of requirements for powertrain driveability study. This article is very interesting to read.